# Selective Interactions of *O*-Methylated Flavonoid Natural Products with Human Monoamine Oxidase-A and -B

**DOI:** 10.3390/molecules25225358

**Published:** 2020-11-17

**Authors:** Narayan D. Chaurasiya, Jacob Midiwo, Pankaj Pandey, Regina N. Bwire, Robert J. Doerksen, Ilias Muhammad, Babu L. Tekwani

**Affiliations:** 1Department of Infectious Diseases, Division of Drug Discovery, Southern Research, Birmingham, AL 35205, USA; nchaurasiya@southernresearch.org; 2National Center for Natural Products Research, Research Institute of Pharmaceutical Sciences, School of Pharmacy, University of Mississippi, Oxford, MS 38677, USA; ppandey@olemiss.edu; 3Department of Chemistry, University of Nairobi, Nairobi P.O. Box 30197-00100, Kenya; jmidiwo@uonbi.ac.ke; 4Department of BioMolecular Sciences, Division of Medicinal Chemistry, Research Institute of Pharmaceutical Sciences, School of Pharmacy, University of Mississippi, Oxford, MS 38677, USA; rjd@olemiss.edu; 5Department of pure and applied Chemistry, Masinde Muliro University of Science and Technology, Kakamega P.O. Box 190-50100, Kenya; rbwire@mmust.ac.ke

**Keywords:** recombinant monoamine oxidase-A, monoamine oxidase-B, neurological disorder, enzyme kinetics, molecular docking, inhibition activity, flavonoid

## Abstract

A set of structurally related *O*-methylated flavonoid natural products isolated from *Senecio roseiflorus* (**1**), *Polygonum senegalense* (**2** and **3**), *Bhaphia macrocalyx* (**4**), *Gardenia ternifolia* (**5**), and *Psiadia punctulata* (**6**) plant species were characterized for their interaction with human monoamine oxidases (MAO-A and -B) in vitro. Compounds **1**, **2**, and **5** showed selective inhibition of MAO-A, while **4** and **6** showed selective inhibition of MAO-B. Compound **3** showed ~2-fold selectivity towards inhibition of MAO-A. Binding of compounds **1**–**3** and **5** with MAO-A, and compounds **3** and **6** with MAO-B was reversible and not time-independent. The analysis of enzyme-inhibition kinetics suggested a reversible-competitive mechanism for inhibition of MAO-A by **1** and **3**, while a partially-reversible mixed-type inhibition by **5**. Similarly, enzyme inhibition-kinetics analysis with compounds **3**, **4**, and **6**, suggested a competitive reversible inhibition of MAO-B. The molecular docking study suggested that **1** selectively interacts with the active-site of human MAO-A near N5 of FAD. The calculated binding free energies of the *O*-methylated flavonoids (**1** and **4**–**6**) and chalcones (**2** and **3**) to MAO-A matched closely with the trend in the experimental IC_50′_s. Analysis of the binding free-energies suggested better interaction of **4** and **6** with MAO-B than with MAO-A. The natural *O*-methylated flavonoid (**1**) with highly potent inhibition (IC_50_ 33 nM; Ki 37.9 nM) and >292 fold selectivity against human MAO-A (vs. MAO-B) provides a new drug lead for the treatment of neurological disorders.

## 1. Introduction

Monoamine oxidases (EC.1.4.3.4; MAO-A and -B) are FAD-dependent enzymes that are responsible for the metabolism of neurotransmitters such as dopamine, adrenaline, serotonin, and noradrenaline, and also for the inactivation of exogenous arylalkyl amines [1,2,3]. Due to their vital role in neurotransmitter metabolism, these enzymes signify attractive drug targets for the pharmacological therapy of neurodegenerative diseases and neurological disorders [4,5,6,7]. Recent efforts toward the development of MAO inhibitors have been focused on selective, reversible MAO-A or MAO-B inhibitors. The identification of MAO inhibitors could be helpful for numerous aspects of new drug discovery. Selective MAO-A inhibitors are effective in the treatment of depression and anxiety [8,9,10,11]. By contrast, MAO-B inhibitors are suitable for the treatment of the neurodegenerative diseases Alzheimer’s disease and Parkinson’s disease [5,7,8,11,12]. Historically, monoamine oxidase inhibitors (MAOI’s) have been used to treat neurological disorders including depression [10]. Presently, MAO-A inhibitors play an important role in the control of neurological disorders, anxiety, and depression, while MAO-B inhibitors could potentially be used as therapeutic agents for Parkinson’s and Alzheimer’s diseases [13]. Pharmacotherapeutic limitations and adverse effects of the currently available MAO inhibitor drugs require the discovery of new MAO inhibitors with selective inhibition profiles and multi-target neuropharmacological profiles [12]. Natural products provide useful sources for MAO inhibitors combined with neuroprotective actions [3,14,15]. Several plant extracts and bioactive natural product metabolites with catecholaminergic neuropharmacological properties [16] have shown promising utility for the treatment of Alzheimer’s disease and Parkinson’s disease [17].

Recent studies from our lab have reported selective inhibition of MAO with flavonoid natural products [18,19,20]. In continuation of these studies on selected classes of flavonoids as well as other related published reports on various flavonoids [21,22], we have selected a series of methoxylated flavones and chalcones from our repository for further evaluation to explore their activities. The structure of chalcone is different from flavone/flavonol/flavanone, but they are biogenetically correlated. However, from our previous studies on flavone/flavanone [18,19], and those reported for chalcones [23] as valid MAO inhibitors, the difference in structures did not explain the structure–activity relationship for the inhibition of MAO A/B. Therefore, we chose to study the MAO inhibitory activity of these two types of flavonoids (flavone/flavonol and chalcone), including their docking studies. Therefore, these studies were further extended to test a set of related *O*-methylated flavonoids isolated from different plants, namely, *Senecio roseiflorus* (3,4′-di-*O*-methylkaempferol; **1**) [24], *Polygonum senegalense* (2′-hydroxy-4′,6′-dimethoxy-chalcone; **2** and 2′,4′-dihydroxy-6′-methoxy-chalcone; **3**) [25], *Bhaphia macrocalyx* (8-demethylsideroxylin; **4**) [26] *Gardenia ternifolia* (4′-*O*-methylkaempferol; **5**) [27], and *Psiadia punctulata* (5,7-dihydroxy-2,3,4,5-tetramethoxyflavone; **6**), [28] for experimental activities against MAO-A and -B. The isolated compounds are all non-polar chalcones (di-*O*-methylated, **2**, and **3**) and flavones exhibiting mono-*O*-methylation (**4** and **5**) or di-*O*-methylation (**1**) or tetra-*O*-methylation (**6**). There was one (**4**) which even showed ring A C-methylation (**4**) (Figure 1). This study was also extended to determine enzyme kinetics and the mechanism of inhibition of the compounds which showed the best IC_50_ values in the recombinant human monoamine oxidases assays (MAO-A and -B). Furthermore, molecular docking simulations were performed to understand the putative binding modes of the best compounds to MAO-A and -B.

## 2. Results

### 2.1. Isolation, Purification, and Characterization of O-Methylated Flavonoids

The *O*-methylated flavonoids reported in this paper were isolated from various plants, using general methods reported earlier [25,27]. Aerial parts (leaves and branches) were dipped in a non-polar solvent for short periods to wash off the exudates into the solvent (without affecting cell vacuole compounds). The solvents used were normally medium polarity solvents such as acetone or ethyl acetate. The solvent was removed using a rotary evaporator and the remaining solid materials were subjected to column chromatography using silica gel as a stationary phase and eluting with hexane/dichloromethane in a gradient fashion continuously increasing polarity followed by dichloromethane/methanol. Compounds **1**–**6** were isolated from different plants, namely, *S. roseiflorus* (**1**) [24], *P. senegalense* (**2** and **3**) [25], *G. ternifolia* (**5**) [27], *P. punctulata* (**6**) [28], and *B. macrocalyx* (**4**). Compound **4** is the first report from the genus *Baphia*. The isolation of compound **4** was not reported in the literature by us, therefore, its structure was determined using 1D and 2D NMR spectral data and TOF-MS (see Material and Methods section for details).

### 2.2. Enzyme Inhibition and Kinetics Mechanism of MAO-A and -B with Compounds **1**–**6**

The inhibition (IC_50_) of the MAO-A and -B enzymes by compounds **1**–**6** are shown in Table 1. Compounds **1**, **2**, and **5** showed selective potent inhibition of MAO-A compared to compound **3**, which was potent at MAO-A but only slightly selective for MAO-A over -B. Compounds **4** and **6** were more potent than **3** at MAO-B and were selective for MAO-B over -A.

Furthermore, the MAO-A inhibition mechanisms of compounds **1**–**3** and **5** were studied, using varying concentrations of kynuramine, a nonselective substrate, to investigate the nature of inhibition of the enzymes. Based on dose–response inhibition results, at least two concentrations of **1**–**3** and **5** were selected for the inhibition kinetics assay—one below and another above the IC_50_ value. Three sets of assays were completed at varying concentrations of the substrate for each experiment, one control without inhibitor and the others with two different concentrations of the inhibitor. The data were evaluated by double reciprocal Lineweaver-Burk plots for determination of the Ki (i.e., inhibition/binding affinity) values. Binding of compounds **1**–**3** and **5**, with human MAO-A, yielded the Km value (i.e., the affinity of the substrate for the enzyme) as well as Vmax (maximum enzyme activity) (Figure 2A–D). Ki values were computed from the double reciprocal plots (Table 2). Binding of compounds **3**, **4**, and **6** to human MAO-B yielded the Km value (i.e., the affinity of the substrate for the enzyme) as well as Vmax (maximum enzyme activity) (Figure 3A–C). Ki values were computed from the double reciprocal plots (Table 2). Compounds **3**, **4**, and **6** showed inhibitory activity of MAO-B with substantially high affinity (Ki = 1.242, 0.809, and 0.874 μM, respectively) (Table 2).

### 2.3. Binding and Time-Dependent Assays of MAO-A and -B with Compounds **1**–**6**

The characteristics of binding of compounds **1**–**3** and **5** with MAO-A were investigated by the equilibrium-dialysis assay. High concentrations of the compounds **1**–**3** and **5** (10.0, 25.0, 25.0, and 100.0 μM, respectively) were incubated with the MAO-A enzyme for 20 min and the resulting enzyme–inhibitor–complex preparation was dialyzed overnight against the 0.025 M phosphate buffer (pH 7.4). The activities of the enzyme were analyzed before and after the dialysis (Figure 4). The binding of compounds **1**–**3** with MAO-A was reversible and compound **5** showed partial reversibility (Table 2). Incubation of MAO-B with compounds **3**, **4**, and **6** (50.0, 50.0, and 50.0 μM, respectively) produced more than 60% inhibition of activity, and 80% of the activity of the enzyme was recovered after dialysis (Figure 5). Thus, the binding of compounds **3**, **4**, and **6** with MAO-B was reversible (Table 2). The selective MAO-B inhibitor deprenyl was confirmed to bind irreversibly with the enzyme (Table 2).

Further investigation of the time dependence of the assay showed that the inhibition of MAO-A by compounds **1**–**3** and **5** was not time-dependent (Figure 6A). The compounds **3**, **4**, and **6** also did not show time-dependent inhibition of MAO-B (Figure 6B). For validation, we have run MAO-A and -B standards simultaneously for the time-dependent assay.

### 2.4. Computational Analysis of Enzyme–Inhibitor Interactions

A molecular docking study was performed to understand the binding pose and interaction profiles of compounds **1**–**6** to MAO-A and -B. Schrödinger’s Induced-Fit docking protocol was adopted to consider the optimal geometry of the protein–ligand complex after conformational changes induced by the bound ligand. The GlideScores and binding free-energies of compounds **1**–**6** in the active sites of the *h*MAO-A and *h*MAO-B X-ray crystal structures are presented in Table 3. The docking protocol used in this study was validated by self- or native-docking. The native ligands, harmine and pioglitazone, were extracted from the X-ray structures of MAO-A and -B, respectively, and docked into their corresponding protein models. The calculated RMSD between the docked and experimental poses were found to be identical <0.6 Å, which verified the suitability of the docking method for the current study. The putative binding mode and interactions of the best compounds with the X-ray crystal structures of MAO-A and -B are presented in Figure 7. The calculated binding free energies vary between 28–76 kcal/mol against MAO-A and -B. Since some of the measured Ki values are in the micromolar range, the binding affinities should be somewhere around 6–10 kcal/mol. This is a known limitation of the employed computations, which are useful not on an absolute scale but in relative terms among structurally similar ligands, which is the focus here Compound **1** exhibited a strong binding affinity to the MAO-A receptor in terms of GlideScore and binding free energy (ΔG = −57.522 kcal/mol) compared to the other compounds. The *p*-methoxy phenyl at the C-2 position (Ring-B) of **1** showed π–π stacking with Phe208 and was surrounded by an array of hydrophobic residues, including Leu97, Phe108, Ala111, Ile180, and Ile325. The hydroxyl at C-5 of ring A formed H-bonding with N5 and C=O of FAD and the hydroxyl at C-7 of ring A exhibited water-mediated H-bonding with Tyr444. In addition, Ring A was surrounded by strong hydrophobic residues Tyr69, Tyr197, Tyr407, and Tyr444. The best GlideScore and binding free energy matched well with the experimental binding affinity of **1.** Interestingly, **1** and **5** had the difference of –OCH_3_ and –OH, respectively, at the C-3 position of ring C but had significant differences in their MAO-A binding affinity. Our docking results also predicted relatively poor GlideScores and binding free energy for compound **5** compared to compound **1.** After careful observation, we have found that the methoxy group at C-3 of **1** exhibited strong hydrophobic interactions with Ile335 and Leu337 compared to the hydroxyl group at C-3 of **5**. Compounds **1** and **5** have very similar poses; however, compound **1** slightly shifted towards FAD (~1.5 Å) compared to **5**. In addition, the hydroxyl at Ring A of **5** did not show any H-bonding with FAD, further helping to explain the poorer binding affinity of compound **5** for MAO-A. Interestingly, compound **1** showed better GlideScore and binding free energy for MAO-B than for MAO-A (see Table 3); however, its best-ranked docking pose left it 15 Å away from the N5 of FAD, which apparently is an unrealistic docking prediction. In the search for an alternative pose for **1** to MAO-B, we found a pose in which **1** fit into the active site of MAO-B (near N5 of FAD) with a GlideScore of −9.705 kcal/mol, and that is the one reported in Table 3. The substitutions of acetyl (-CH_3_CO) and methylsulfone (-SO_2_CH_3_) at the C-3 and C-4′ of Ring B of **1** are predicted to enhance the affinity towards MAO-A. The poly-substituted methoxy group at Ring B caused a loss of binding affinity towards MAO-A but submicromolar activity towards MAO-B.

The next structural category, chalcone, represented by compounds **2** and **3**, was analyzed. Compound **2** showed the more negative binding free energy (ΔG = −47.724 kcal/mol) compared to compound **3** (ΔG = −37.683 kcal/mol) for binding to MAO-A, and these data match closely with the experimental binding affinities (cf. Table 1). The only structural difference between compounds **2** and **3** involves C-4′ carrying methoxy and hydroxyl moieties, respectively. The docked pose of **2** in the MAO-A receptor showed H-bonding of its hydroxyl moiety at C-6′ and its C-1 carbonyl (water-mediated H-bonding) with Asn181. In addition, the oxygen of the methoxy at C-4′ exhibited water-mediated H-bonding with Gln215 and Tyr444. The major difference of binding free energy between **2** and **3** was because of an additional strong hydrophobic interaction (CH…C, C…C, and CH…π) of the C-4′ methoxy group of **2** with Tyr69, Phe352, and Tyr407, respectively.

The GlideScores and binding free energies of the flavonoids **4** and **6** showed a better binding affinity for interaction with MAO-B (**4**: GlideScore = −10.225 kcal/mol, ΔG = −53.574 kcal/mol and **6**: GlideScore = −11.191 kcal/mol, ΔG = −68.053 kcal/mol) than with MAO-A. Compounds **4** and **6** docked in a very similar orientation and showed H-bonding interactions between their C-4 carbonyl and Cys172. In addition, the C-5 hydroxyl of **4** and **6** showed water-mediated H-bonding with Tyr188, and Gln206 and the C-4 hydroxyl had direct hydrogen-bonding with the backbone carbonyl of Cys172. Ring A had an orientation towards the isoalloxazine ring of FAD and was surrounded by an array of hydrophobic residues, including Tyr60, Phe343, Tyr398, and Tyr435. Furthermore, Ring B was surrounded by hydrophobic residues Leu164, Leu167, Phe168, Ile199, Ile316, and Tyr326 (including π–π stacking for Tyr326). Overall, the docking results of compounds **1**–**6** were in good agreement with the experimental binding data for MAO-A and -B.

## 3. Discussion

The molecules with reversible selective inhibition of MAO-A or MAO-B have therapeutic potential for the treatment of neurological and psychiatric disorders, especially caused due to depletion of neurotransmitter biogenic amines [9,29,30]. Previous studies from our lab have reported selective inhibition of human MAO-B with flavonoid natural products [18,19,20]. A recent study has also reported MAO-A and MAO-B inhibition activity by acacetin 7-*O*-(6-*O*-malonylglucoside), a derivative of acacetin isolated from *Agastache rugosa* plant leaves [31]. The follow-up studies presented here with a select set of *O*-methylated flavonoids (**1**–**6**) identified MAO inhibitors selective against both MAO-A (**1**–**3** and **5**) and MAO-B (**4** and **6**). Compounds **1**–**3** interact with MAO-A through reversible binding as assessed by the enzyme–inhibitor complex equilibrium dialysis assay, while the binding of compound **5** with MAO-A was partially reversible. The inhibition of MAO-A by compounds **1**–**3** and **5** was not time-dependent. A recent paper reported MAO-A and -B inhibition activity by natural constituent acacetin 7-*O*-(6-*O*-malonylglucoside) that was isolated and purified from *Agastache rugosa* plant leaves [32]. Compounds **3**, **4**, and **6** also interact with MAO-B reversibly, as assessed by the enzyme–inhibitor complex equilibrium dialysis assay, and the inhibition was not time-dependent.

Computational analysis of the binding of **1**–**6** with human MAO-A and -B revealed the putative binding mode and interaction profiles of the compounds with MAO-A and -B. Among all the *O*-methylated flavonoids, **1** showed the strongest computed interaction with MAO-A and exhibited H-bonding with N5 and C=O of FAD through the hydroxyl at C-5 of Ring A. In addition, the hydroxyl at C-7 of Ring A also exhibited water-mediated H-bonding with Tyr444. **1** also showed π–π interactions with Phe208 and was surrounded by an array of hydrophobic residues. On the other hand, **4** and **6** showed strong interactions with MAO-B and shared an identical binding mode with MAO-B. This study suggests that it would be worthwhile to perform further evaluation of compounds **1**–**6** including considering the effects of their MAO-A and -B inhibitory actions in experimental animal models of neurological and/or neurodegenerative disorders.

The *O*-methylated flavonoids are predominant bioactive secondary metabolites present in several plants [32]. The *O*-methylated flavonoids are generated in plants through the action of specific *O*-methyltransferase (OMT) enzymes [33]. *O*-methylation changes the solubility of flavonoids and improves bioactive properties compared to their non-methylated counterparts [34]. The natural product *O*-methylated flavonoid 3,4′-di-*O*-methylkaempferol (**1**) isolated from *S. roseiflorus* was identified as a highly potent inhibitor of human MAO-A with IC_50_ and Ki values of 33 nM and 37.8 nM, respectively. The metabolite **1** was more than 292-fold selective for MAO-A over MAO-B. The compound formed a reversible enzyme–inhibitor complex and was had a very low Ki for MAO-A. With its highly potent MAO-A inhibition and extraordinary selectivity for human MAO-A over MAO-B, **1** is worth optimizing further as a new-drug lead and merits advancement to preclinical evaluations regarding utility for the treatment of neurological and psychiatric disorders.

## 4. Materials and Methods

### 4.1. Enzymes

Recombinant human monoamine oxidase (rhMAO-A and -B) enzymes were purchased from BD Biosciences (Bedford, MA, USA). Kynuramine, clorgyline, deprenyl, and DMSO were obtained from Sigma Chemical (St Louis, MO, USA).

### 4.2. Isolation and Identification of Compounds **1**–***6***

Compounds **1**–**3**, **5**, and **6** (Figure 1) were isolated and characterized by ^1^H and ^13^C-NMR spectra from *Senecio roseiflorus* (**1**), *Polygonum senegalense* (**2** and **3**), *Gardenia ternifolia* (**5**), and *Psiadia punctulata* (**6**) plant species collected from Kenya [24,25,27,28] (*vide supra*). In addition, compound **4** was isolated from the leaves of *B. macrocalyx*, collected from Southern Tanzania (Mikindaniya Leo village). A voucher specimen (FMM 3579) was deposited at the Herbarium of the Department of Botany, University of Dar es Salaam, Tanzania. The dried pulverized plant material (5 kg) underwent sequential cold solvent extraction with hexane, EtOAc, and finally MeOH, each soaked in each solvent for 24 h, then the solvents were evaporated with a rotavap yielding hexane, EtOAc, and MeOH extracts (100, 300 and 200 g, respectively). Column chromatography of EtOAc extract (70 g) on silica gel (Merck silica gel 60; 0.40–0.63 mm) eluting with a hexane-DCM gradient, from 100% hexane to 100% DCM and finally in methanol afforded 10 fractions. The fractions eluting with hexane: DCM (3:7) afforded compounds **4** (8-desmethylsideroxylin, 13.2 mg); obtained as yellow needles, mp 275–277 °C (Lit. 275–277 °C; ESI MS (TOF +ve, Finnigan Mat SSQ 7000): *m/z* 299.1 ([M+H]^+^, C_17_H_14_O_5_+H) ^1^H-NMR (600 MHz, Avance Bruker): 6.82, s, 1H (H-3), 6.82, s, 1H (H-8), 7.95, *dd*, 2H, J = 2.4, 9.0 Hz (H-2′/6′), 6.93, *dd*, 2H, J = 2.4, 9.0 Hz (H-3′/5′), 1.98, *s*, 3H (C-6-Me), 3.90, *s*, 3H (C-7-OMe), 13.08, *s*, 1H (C-5-OH), ^1^H-NMR spectra was in agreement with those reported in the literature); ^13^C-NMR: 161.3 (C-2), 103.1 (C-3), 182.0 (C-4), 104.4 (C-4a), 157.5 (C-5), 107.6 (C-6), 163.1 (C-7), 90.4 (C-8), 155.5 (C-8a), 121.2 (C-1′), 128.6 (C-2′/6′), 116.2 (C-3′/5′),163.9 (C-4′), 7.4 (CH_3_ at C-6), 56.3 (OCH_3_ at C-7). All NMR spectra of **1**–**6** are provided in Appendix A.

### 4.3. MAO Inhibition Assay

In this study, we have investigated the effect of the isolated constituents (**1**–**6**) on human recombinant MAO-A and -B. The kynuramine oxidation deamination assay was performed in 96-well plates as previously reported, with modification [18,35]. A fixed concentration of kynuramine substrate and varying concentrations of test compounds or inhibitor were used to determine the IC_50_ values. Kynuramine concentrations for MAO-A and -B were 80 μM and 50 μM, respectively. The concentrations of compounds **1**–**6** varied from 0.001 μM to 100 μM for the rhMAO-A and -B enzyme activity inhibition. The test compounds **1**–**6** were dissolved in DMSO, diluted in the buffer solution just before the assay, and pre-incubated with the enzyme for 10 min at 37 °C. The final concentration of DMSO in the enzyme-assay reaction mixtures did not exceed 1%. The enzymatic reactions were initiated by the addition of MAO-A (5 μg/mL) or -B (12.5 μg/mL), and incubated for 20 min at 37 °C. The enzyme reactions were terminated by the addition of 78 μL of 2N NaOH. The formation of 4-hydroxyquinoline (the enzyme reaction end product) was recorded fluorometrically on a SpectraMax M5 fluorescence plate reader (Molecular Devices, Sunnyvale, CA, USA) with an excitation (320 nm) and emission (380 nm) wavelength, using the Soft MaxPro-6 program. The inhibition effects of enzyme activity were calculated as the percent of product formation compared to the corresponding controls (enzyme–substrate reaction) without inhibitors. The assay controls, to define the interference of the test compounds with the fluorescence measurements, were set up simultaneously, and the enzyme or the substrate was added after stopping the reaction.

### 4.4. Determination of IC_50_ Values

The enzyme assays were performed at a fixed concentration of the substrate kynuramine (80 μM for MAO-A and 50 μM for MAO-B) and different concentrations of the test compounds (**1**–**6**). The dose–response enzyme-inhibition curves were generated using Microsoft^®^ Excel and the IC_50_ values were computed with XLfit^®^.

### 4.5. Enzyme Kinetics and Mechanism Studies

For determination of the binding affinity of the inhibitor (Ki) to MAO-A and -B, the enzyme assays were carried out at different concentrations of kynuramine substrate (1.90 μM to 500 μM) and varying concentrations of the inhibitors/compound. The flavonoids (**1**–**6**) were tested at 0.030–0.100 μM for MAO-A and 0.100–0.500 μM for -B. The controls without inhibitor were also run simultaneously. The results were analyzed by SigmaPlot version 10 using standard double reciprocal Lineweaver-Burk plots for computing Km and Vmax values, which were further analyzed to determine the Ki values [18,36,37].

### 4.6. Analysis of Binding of Inhibitor with The Enzymes

Enzyme-inhibitors mostly produce inhibition of the target enzyme through the formation of an enzyme–inhibitor complex. The formation of the enzyme–inhibitor complex may be accelerated in the presence of a high concentration of the test inhibitor. The property of binding of test compounds to MAO-A or -B was determined by the formation of the enzyme–inhibitor complex by incubation of the enzyme with a high concentration of the test compound. This was followed by extensive equilibrium dialysis of the enzyme–inhibitor complex. Recovery of catalytic activity of MAO-A and -B was determined before and after the dialysis. The MAO-A enzyme (0.2 mg/mL protein) was incubated with each test compound: **1** (10.0 μM), **2** (25.0 μM), **3** (25.0 μM), **5** (100.0 μM) and clorgyline (0.100 μM), in 1 mL of potassium phosphate buffer (100 mM, pH 7.4). After 20 min incubation at 37 °C, the reaction was stopped by chilling the tubes in an ice bath. Similarly, the MAO-B enzyme (0.2 mg/mL protein) was incubated with each test compound: **3** (50.0 μM), **4** (50.0 μM), **6** (50.0 μM), and deprenyl (0.500 μM), in 1.0 mL potassium phosphate buffer (100 mM, pH 7.4). After 20 min incubation at 37 °C, the reaction was stopped by chilling the tubes in an ice bath. All the samples with enzyme–inhibitor complex were individually dialyzed against potassium phosphate buffer (25 mM; pH 7.4) at 4 °C for 16–18 h (including three buffer changes). The control enzyme (without inhibitor) was also run through the same procedure and the activity of the enzyme was determined before and after the dialysis [36].

### 4.7. Time-Dependent Inhibition of the Enzyme

To investigate if the binding of the inhibitor with MAO-A and -B followed time-dependent inhibition kinetics, the enzyme was pre-incubated with the inhibitor for different time periods (0–15 min). The compound concentrations used to test time-dependent inhibition were: **1** (0.20 μM), **2** (1.6 μM), **3** (3.0 μM) and **4** (16.0 μM) and clorgyline (0.010 μM), with MAO-A (5.0 μg/mL). The inhibitor concentrations used to test time-dependent inhibition were: **3** (3.0 μM), **4** (1.0 μM), **6** (2.0 μM), and deprenyl (0.070 μM), with MAO-B (12.5 μg/mL). The controls without inhibitors were also run simultaneously. The activities of the MAO-A and -B enzymes were determined as described above and the percentage of enzyme activity remaining was plotted against the pre-incubation time to determine time-dependent inhibition.

### 4.8. Computational Analysis of The Interaction of Test Compounds with MAO-A or -B

The X-ray crystal structures of MAO-A (PDB ID: 2Z5X) and MAO-B (PDB ID: 4A79 [38] were directly imported from the Protein Data Bank website (https://www.rcsb.org) to Maestro [39] using the Protein Preparation Wizard module of the Schrödinger software (Cambridge, MA, USA) [40]. We followed a similar method and protocol for docking as previously described [19]. The protein structures of MAO-A and -B were each used as monomers in the docking study. In brief, these proteins were prepared by adding hydrogens, adjusting bond orders, adding missing side chains, setting the proper ionization states at pH 7.4, refining overlapping atoms, and making H-bond assignments using PROPKA at pH 7.0. The water molecules beyond 5 Å from the co-crystallized ligands were deleted and the protonation states of the co-crystallized ligands were generated using Epik at pH 7.4. During the refinement process, water molecules with fewer than two H-bonds to non-waters were also removed and, finally, restrained minimization of hydrogens only was performed using the Optimized Potentials for Liquid Simulations 3 (OPLS3) force field [41]. The cofactor FAD was not removed during protein preparation and docking. The 2D structures of compounds **1**–**6** were sketched in the 2D sketcher module of Maestro, prepared, and energy-minimized at a physiological pH of 7.4 using the LigPrep module [42] of the Schrödinger software. The compounds were docked as neutral molecules. The OPLS3 force field was used for protein and ligand preparation, and docking. The active sites of the MAO-A and -B proteins were generated using the centroid of the co-crystallized ligands of 2Z5X and 4A79, respectively. The Induced Fit docking [43] protocol was used for the docking of compounds **1**–**6.** The standard precision (SP) docking method was applied during the initial docking stages. In the initial Glide docking, the receptor and the ligand were “softened” by van der Waals radii scaling. The scaling factor was chosen to be 0.50 for both the ligand and the receptor to permit enough flexibility for the ligand to dock in the best poses. The “trim-side chains” option was not used in this study. The maximum number of poses was chosen to be 20. In the next step, residues that are within 5 Å of the active site (ligand) were refined using the “Prime Refinement” Table In the final step, a threshold of 30 kcal/mol was used to redock the best structure, for eliminating high-energy structures from the Prime refinement step. The top 20 poses were kept for analysis and the best poses were selected based on IFD scores and visual inspection of protein–ligand interactions. The best docking poses were subjected to binding free-energy calculations using the Prime MM-GBSA module of the Schrödinger software allowing protein flexibility within 5 Å of the ligand. Only protein side-chains were minimized during the calculations. Finally, the Maestro Version 11.5 molecular graphics system was used to create all the computationally derived figures.

## 5. Conclusions

Screening of a selected set of *O*-methylated flavonoid constituents isolated from *Senecio roseiflorus*, *Bhaphia macrocalyx*, *Polygonum sengalense*, *Psiadia punctulata*, and *Gardenia ternifolia* identified compounds **1**–**3** and **5** as potent and selective inhibitors of human MAO-A, relative to MAO-B, and compounds **4** and **6** as selective inhibitors of human MAO-B. Further investigations suggested compounds **1**–**3** as reversible and competitive inhibitors and compound **5** as a partially reversible mixed-type inhibitor of MAO-A and compounds **3**, **4**, and **6** as reversible and competitive inhibitors of MAO-B. The computational results for compounds **1**–**6** were in good agreement with the experimental binding data for MAO-A and -B. The compounds 1 and 6 with high potency and selectivity of inhibition against MAO-A and MAO-B, respectively, may be promising new drug leads for further development as therapeutic treatment of neurological disorders, depression, Alzheimer’s disease, and Parkinson’s disease. It is important to mention that the flavonoid scaffold possesses promiscuous biological activity that may be due to inherent structural features. For this reason, they should be treated with caution as lead compounds for drug development.

## Figures and Tables

**Figure 1 molecules-25-05358-f001:**
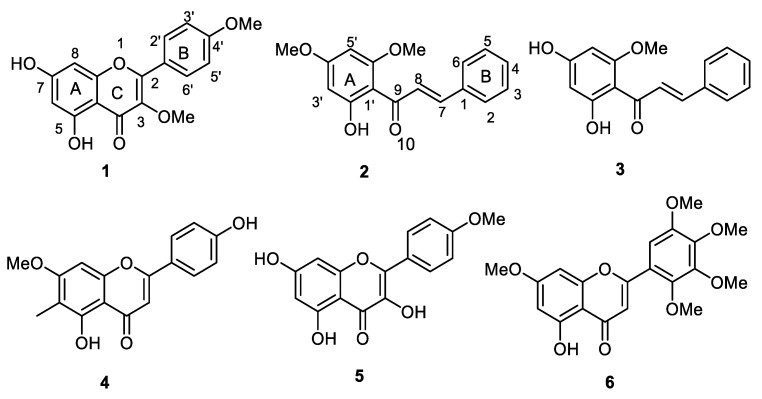
Chemical structures of *O*-methylated flavonoids **1**–**6**.

**Figure 2 molecules-25-05358-f002:**
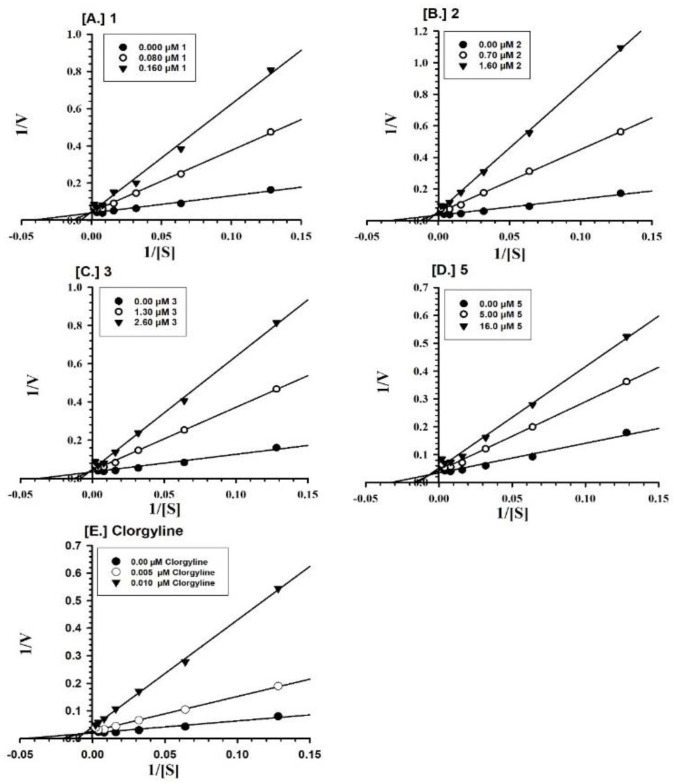
Kinetics characteristics of the inhibition mechanism of recombinant human MAO-A by compounds (**A**) **1**; (**B**) **2**; (**C**) **3**; (**D**) **5**; and (E) Clorgyline. Each point shows the mean value of three observations.

**Figure 3 molecules-25-05358-f003:**
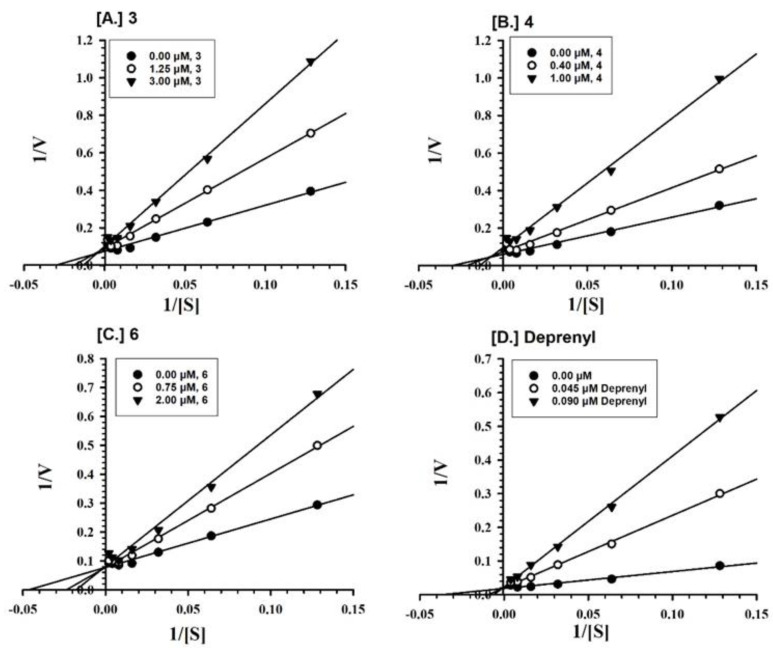
Kinetics characteristics of the inhibition mechanism of recombinant human MAO-B by compounds (**A**) **3**; (**B**) **4**; (**C**) **6**; and (**D**) l-Deprenyl. Each point shows the mean value of three observations

**Figure 4 molecules-25-05358-f004:**
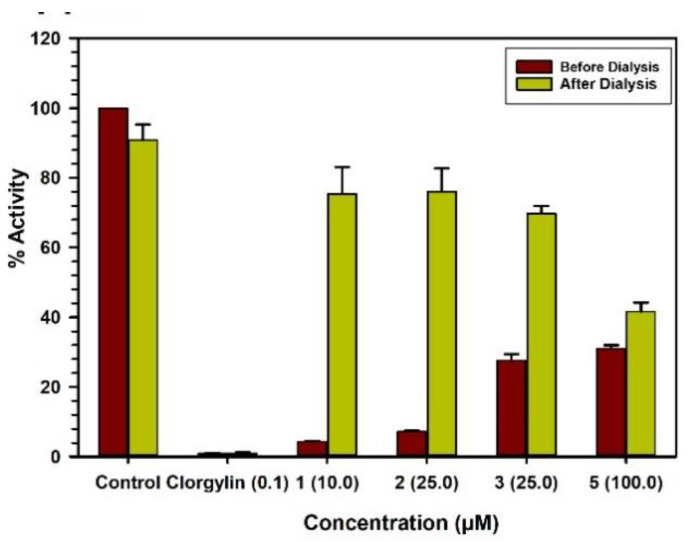
Binding modes of compounds **1** (10.0 μM), **2** (25.0 μM), **3** (25.0 μM), **5** (100.0 μM) and clorgyline (0.100 μM) with MAO-A.

**Figure 5 molecules-25-05358-f005:**
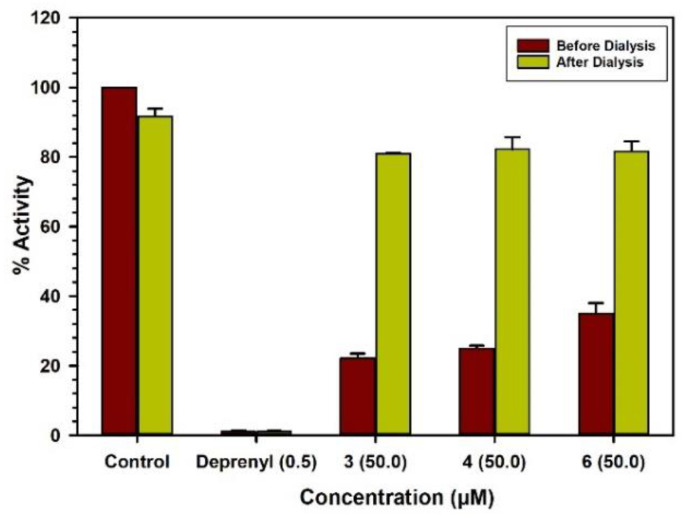
Binding modes of compounds **3** (50.0 μM), **4** (50.0 μM), **6** (50.0 μM) and l-deprenyl (0.500 μM) with MAO-B.

**Figure 6 molecules-25-05358-f006:**
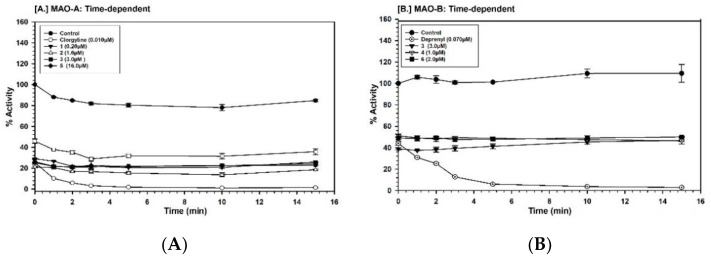
(**A**) Time-dependent inhibition of recombinant human MAO-A by compounds **1** (0.20 μM), **2** (1.6 μM), **3** (3.0 μM), **5** (16.0 μM) and l-deprenyl (0.010 μM). The remaining activity was expressed as % of initial activity. Each point represents the mean ± S.D. of triplicate values. (**B**) Time-dependent inhibition of recombinant human MAO-B by compounds **3** (3.0 μM), **4** (1.0 μM), **6** (2.0 μM) and l-deprenyl (0.070 μM). The remaining activity was expressed as % of the initial activity. Each point represents the mean ± S.D. of triplicate values.

**Figure 7 molecules-25-05358-f007:**
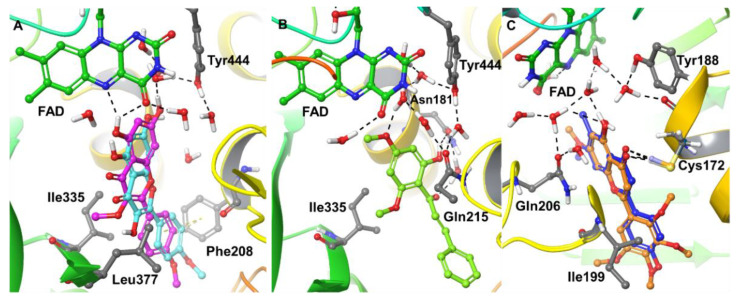
Three-dimensional (3D) representation of the protein–ligand interactions of **1**–**6** with the X-ray crystal structures of MAO-A and -B. (**A**) **1** (C magenta, stick model) and **5** (C cyan, stick model) with MAO-A, (**B**) **2** (C light green, stick model) with MAO-A, (**C**) **4** (C blue, stick model) and **6** (C orange, stick model) with MAO-B. Some crystallographic waters (O red, H white, stick model), FAD (C dark green), and the important residues of MAO-A and MAO-B (C gray) are also shown. The black dashed lines represent H-bonding.

**Table 1 molecules-25-05358-t001:** Inhibition of recombinant human MAO-A and MAO-B by a selected set of *O*-methylated flavonoids **1**–**6**. The results with significantly potent inhibition are presented in bold.

Compounds	MAO-A IC_50_ (µM) ^a^	MAO-B IC_50_ (µM) ^a^	SI ^b^	SI ^c^
**1**	**0.033 ± 0.042**	9.667 ± 2.309	0.0034	**292.93**
**2**	**0.407 ± 0.075**	5.933 ± 0.833	0.082	**14.57**
**3**	**1.167 ± 0.513**	2.700 ± 0.794	0.432	2.31
**4**	5.167 ± 1.106	**0.800 ± 0.180**	**6.451**	0.154
**5**	**1.350 ± 0.198**	>100	-	>74.07
**6**	87.501 ± 3.536	**0.875 ± 0.035**	**100.0**	0.009
Clorgyline ^b^	0.0065 ± 0.001	-	-	-
Deprenyl ^c^	-	0.043 ± 0.011	-	-

^a^ The IC_50_ values computed from the dose–response inhibition curves are mean ± S.D. of at least triplicate observations. ^b^ SI Selectivity index: MAO-A IC_50_/MAO-B IC_50_. ^c^ Selectivity Index: MAO-A IC_50_ / MAO-B IC_50_. ^b^Clorgyline and ^c^L-deprenyl were used as positive controls for MAO-A and -B, respectively.

**Table 2 molecules-25-05358-t002:** Inhibition/binding affinity constants (Ki) for inhibition of recombinant human MAO-A by compounds **1**–**3** and **5** and of MAO-B by compounds **3**, **4**, and **6.**

Compounds	Monoamine Oxidase-A		Monoamine Oxidase-B	
	Ki (μM) ^a^	Type of Inhibition	Ki (μM) ^a^	Type of Inhibition
**1**	0.0379 ± 0.0008	Competitive/Reversible	-	-
**2**	0.339 ± 0.219	Competitive/Reversible	-	-
**3**	0.633 ± 0.107	Competitive/Reversible	1.242 ± 0.600	Competitive/Reversible
**4**	-	-	0.809 ± 0.093	Mixed/Reversible
**5**	3.531 ± 0.265	Mixed/Partially Reversible	-	-
**6**	-	-	0.874 ± 0.069	Competitive/Reversible
Clorgyline ^b^	0.0018 ± 0.0003	Mixed/Irreversible	-	-
Deprenyl ^b^	-	-	0.0101 ± 0.0034	Mixed/Irreversible

^a^ The results are presented as the mean ± SD of three observations; ^b^ Clorgyline and l-deprenyl were used as positive controls for MAO-A and -B, respectively.

**Table 3 molecules-25-05358-t003:** GlideScores and binding free energies of compounds **1**–**6** to MAO-A and -B.

Compounds	Experimental IC_50_ (µM) ^a^		GlideScore (kcal/mol)		Binding Free-Energy (kcal/mol)	
	MAO-A	MAO-B	MAO-A	MAO-B	MAO-A	MAO-B
**1**	0.033 ± 0.04	9.667 ± 2.309	−11.667	−10.028 ^b^	−57.522	−76.353 ^b^
**2**	0.407 ± 0.075	5.933 ± 0.833	−10.686	−9.999	−47.724	−28.119
**3**	1.167 ± 0.513	2.700 ± 0.794	−9.951	−10.579	−37.683	−51.309
**4**	5.167 ± 1.106	0.800 ± 0.180	−9.664	−10.225	−35.043	−53.574
**5**	1.350 ± 0.198	>100	−10.567	ND ^c^	−47.035	ND ^c^
**6**	87.501 ± 3.536	0.875 ± 0.035	−7.239	−11.191	−34.651	−68.053

^a^ The data are mean ± SD of three observations.; ^b^ The best pose is 15 Å away from the N5 of FAD (substrate active site); the numbers given here are for an alternate pose (see text); ^c^ ND = Not determined.

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
