# Peer review of "Selective Interactions of O-Methylated Flavonoid Natural Products with Human Monoamine Oxidase-A and -B"

_molecules, 2020, doi:10.3390/molecules25225358_

Round 1

Reviewer 1 Report

The authors of the manuscript " Selective interactions of O-methylated flavonoid natural products with human monoamine oxidase-A and -B” The authors described the Monoamine oxidase of a series of flavonoids, isolated from different medicinal plants from Nigeria.

Although the manuscript is well written, the biological analysis was well carried out, and the results are interesting, however there is a lack of explanation in several parts.

For example: The abstract is too long and maybe more concise. In the introduction the authors failed in explain what they want to do with these several different flavonoids a comparison or structure activity relationship or just evaluate them. In the introduction the authors must be more detailed with the type of flavonoid a chalcone is way different than a flavonol or flavone. Which will be the implications? At this point seems like they gathered some compounds and tested but no real point. In the discussion the authors must compare the bioactive compounds, number 1 for MAO A and number 4 and 6 for MAO B with reported data and build from there. Which kind of groups are the really important in the kind of structure studied? they can also expand a little and compare computationally and even predict what kind of structural features are important. I also disappointed with the description of the isolation process, please more specific or just use a reference. What about the purity of the compounds? Did the author have NMR? If yes, please include the spectral data as a supporting? The use of the chemical descriptors must be following the conventions. For example, the O-methylated, the O must the italics.

Overall, the manuscript needs to improve and could be accepted after major revisions

Author Response

Response to Reviewer 1 Comments

The authors of the manuscript " Selective interactions of O-methylated flavonoid natural products with human monoamine oxidase-A and -B” The authors described the Monoamine oxidase of a series of flavonoids, isolated from different medicinal plants from Nigeria.

Point 1: Although the manuscript is well written, the biological analysis was well carried out, and the results are interesting, however there is a lack of explanation in several parts.

For example: The abstract is too long and maybe more concise.

Response 1: The abstract has been revised to reflect the essence and only key points of the study.

 All compounds reported in the paper were isolated from Kenyan flora. We have not mentioned anywhere in the manuscript that these compounds are from “Nigerian flora”.

In addition, we now have included a sentence stating that all compounds were isolated from Kenyan flora, namely.

Point 2: In the introduction the authors failed in explain what they want to do with these several different flavonoids a comparison or structure activity relationship or just evaluate them.

Response 2:  We have clarified the purpose of the work.

 Point 3: In the introduction the authors must be more detailed with the type of flavonoid a chalcone is way different than a flavonol or flavone. Which will be the implications? At this point seems like they gathered some compounds and tested but no real point.

Response 3: The introduction has been revised to addressed this-

In continuation to our earlier studies on MAO activity on selected classes of flavonoids (*REF 1, 2), as well as related published reports on various flavonoids (new/ existing REFS), we have selected a series of methoxylated flavones and chalcones from our repository for further evaluation to explore their activities.  Yes, the structure of chalcone is different than flavone/ flavonol/ flavanone, but they are biogenetically correlated. However, from our previous studies on flavone/ flavanone (*REF 1, 2), and those reported for chalcones (*REF 3) as valid MAO inhibitors, the difference in structures could not draw a solid foundation to conclude their MAO A/B SAR. Therefore, we choose to study the MAO inhibitory activity of these two types of flavonoids (flavone/ flavonol and chalcone), including their docking studies, if indeed there are any potential selectivity in their activities.

* REF 1- Monoamine Oxidase Inhibitory Constituents of Propolis: Kinetics and Mechanism of Inhibition of Recombinant Human MAO-A and MAO-B.  Chaurasiya ND, Ibrahim MA, Muhammad I, Walker LA, Tekwani BL. Monoamine Oxidase Inhibitory Constituents of Propolis: Molecules. 19 (11):18936-52, 2014.

* REF 2- Selective Inhibition of Human Monoamine Oxidase B by Acacetin 7-Methyl Ether Isolated from Turnera diffusa(Damiana). Narayan D. Chaurasiya, Jianping Zhao, Pankaj Pandey, Robert J. Doerksen, Ilias Muhammad, Babu L. Tekwani. Molecules 24, 810, 2019; doi:10.3390/molecules24040810.

* REF 3- Franco Chimenti, Rossella Fioravanti, Adriana Bolasco, Paola Chimenti, Daniela Secci, Francesca Rossi,Matilde Ya ́n ̃ez, Francisco Orallo, Francesco Ortuso, and Stefano Alcaro. Chalcones: A Valid Scaffold for Monoamine Oxidases Inhibitors. J. Med. Chem. 2009, 52, 2818–2824.

 Point 4:  In the discussion the authors must compare the bioactive compounds, number 1 for MAO A and number 4 and 6 for MAO B with reported data and build from there.

Response 4: We have added in additional comparison with reported results in the literature. Compound 1 is comparable to galangin (see REF 1), and 4 and 6 to 7-O-methyl acacetin (see REF 2). In addition, compound 4 has a unique with C-7 methyl group.

Point 5: Which kind of groups are the really important in the kind of structure studied? they can also expand a little and compare computationally and even predict what kind of structural features are important.

Response 5: We have already described the structural features of the studied compounds in Section 2.4. Computational analysis of enzyme–inhibitor interactions. We added the following sentence in the revised manuscript: “The substitutions of acetyl (-CH3CO) and methylsulfone (-SO2CH3) at the C-3 and C-4’ of Ring B of 1 are predicted to enhance the affinity towards MAO-A. The poly-substituted methoxy group at Ring B caused a loss of binding affinity towards MAO-A but submicromolar activity towards MAO-B”.

Point 6: I also disappointed with the description of the isolation process, please more specific or just use a reference.

Response 6: All the compounds reported in this manuscript are well known to the literatures reported by our Co-author, Dr. Midiwo’s group using Kenyan flora. Most of these compounds have reported from many other plants as well. The isolation procedures for 1-3, 5 and 6, and their structure elucidation by spectral data were published previously, and supported by references (REF cited 21-24) that we have provided in this manuscript. However, we have now added the isolation method and spectral data of compound 4 (in the Materials and Methods section) which has not been by reported by us previously.

Point 7: What about the purity of the compounds? Did the author have NMR? If yes, please include the spectral data as a supporting?

Response 7: We have added the NMR spectra of all compounds as Supporting Information. Based on the NMR spectra, all tested compounds appear to be pure in the range of 95-97%.

Point 8: The use of the chemical descriptors must be following the conventions. For example, the O-methylated, the O must the italics.

Response 8: Corrected

Reviewer 2 Report

General Comments.

In this manuscript, the author reported the discovery and biological evaluation of 6 O-methylated flavonoids as MAO inhibitors. However, the flavonoids as selective MAO inhibitors have been extensively studied. For example, reference [1,2] listed below and reference 18-20, 27 in the manuscript have reported the results of flavonoids as the selective MAO inhibitors. The molecular structures reported in this manuscript were highly similar to previously reported MAO inhibitors. Therefore, the present work does not appear so original, even though the experiments were well-designed. Moreover, while the potency of discovered MAO-A inhibitor (compound 1) was in the nanomolar range, the subtype selectivity is not good enough, since many inhibitors with SI > 1000 have been reported. For these reasons, I suggest to accept the paper with major revisions.

Detail Comments.

  1. More information on flavonoids as MAO inhibitors should be reviewed in the introduction part. Since the compounds discovered in this study provided less new clues for the design of novel MAO inhibitors, the SAR, i.e., effect of different substituents on flavonoid scaffold against MAO-A/B, should be provided to support the finding in the manuscript.
  2. The detailed parameters of induced-fit docking should be given. For example, which residues were selected to mutate to alanine for initial docking.
  3. The statement in the abstract that MAO-A formed strong and stable interactions with each of the O-methylated flavonoids (2-5) was conflicted with the experimental results of compound.
  4. And, in my opinion, even the flavonoid scaffold was not list as a frequent hit or PAINS compound, they were reported existing in many kinds of plants extensively and to be active in many, many biological assays. They should be treated with special cautions, especially as lead compound for drug development.
  5. The manuscript should be edited carefully.
  6. In Figure 1, the illustration of ring A, B and C mentioned in the paper should be provided.
  7. Table 1 should be well prepared. For example, in the annotation of Table 1, it would be better if “b SI = [selectivity index]” was changed to “b SI means selectivity index, which was defined as MAO-A IC50/MAO-B IC50”. The MAO-B IC50 of compound 5 does not need to be bold. The symbol for Clorgyline and Deprenyl can be change to “c” since “b” was used for selectivity index.
  8. The first line was not indented in part ‘4.2 MAO Inhibitory Assay’. The first sentence in ‘4.7 Computational analysis’ lacked a closing bracket. The word ‘manuscript’ in the last sentence of Author contributions didn’t need to be bold. The journal name of Reference 21 should be abbreviated.

Reference

  1. Jalili-Baleh, L.; Babaei, E.; Abdpour, S.; Nasir Abbas Bukhari, S.; Foroumadi, A.; Ramazani, A.; Sharifzadeh, M.; Abdollahi, M.; Khoobi, M. A review on flavonoid-based scaffolds as multi-target-directed ligands (MTDLs) for Alzheimer's disease. Eur J Med Chem 2018, 152, 570-589, doi:https://doi.org/10.1016/j.ejmech.2018.05.004.
  2. Larit, F.; Elokely, K.M.; Chaurasiya, N.D.; Benyahia, S.; Nael, M.A.; León, F.; Abu-Darwish, M.S.; Efferth, T.; Wang, Y.-H.; Belouahem-Abed, D., et al. Inhibition of human monoamine oxidase A and B by flavonoids isolated from two Algerian medicinal plants. Phytomedicine 2018, 40, 27-36, doi:10.1016/j.phymed.2017.12.032.

Author Response

Response to Reviewer 2 Comments

In this manuscript, the author reported the discovery and biological evaluation of 6 O-methylated flavonoids as MAO inhibitors. However, the flavonoids as selective MAO inhibitors have been extensively studied. For example, reference [1,2] listed below and reference 18-20, 27 in the manuscript have reported the results of flavonoids as the selective MAO inhibitors. The molecular structures reported in this manuscript were highly similar to previously reported MAO inhibitors. Therefore, the present work does not appear so original, even though the experiments were well-designed. Moreover, while the potency of discovered MAO-A inhibitor (compound 1) was in the nanomolar range, the subtype selectivity is not good enough, since many inhibitors with SI > 1000 have been reported. For these reasons, I suggest to accept the paper with major revisions.

Detail Comments.

Point 1: More information on flavonoids as MAO inhibitors should be reviewed in the introduction part. Since the compounds discovered in this study provided less new clues for the design of novel MAO inhibitors, the SAR, i.e., effect of different substituents on flavonoid scaffold against MAO-A/B, should be provided to support the finding in the manuscript.

Response 1: We have added in more information regarding the known SAR of flavonoids as MAO inhibitors. We added in the suggested references.

Point 2: The detailed parameters of induced-fit docking should be given. For example, which residues were selected to mutate to alanine for initial docking.

Response 2: More information has been added to address this concern in the revised manuscript. We added the following to Materials and Methods Section 4.7:

In the initial Glide docking, the receptor and the ligand were “softened” by van der Waals radii scaling. The scaling factor was chosen to be 0.50 for both the ligand and the receptor to permit enough flexibility for the ligand to dock in the best poses. The “trim-side chains” option was not used in this study. The maximum number of poses was chosen to be 20. In the next step, residues that are within 5 Å of the active site (ligand) were refined using the “Prime Refinement” tab. In the final step, a threshold of 30 kcal/mol was used to redock the best structure, for eliminating high-energy structures from the Prime refinement step

Point 3: The statement in the abstract that MAO-A formed strong and stable interactions with each of the O-methylated flavonoids (2-5) was conflicted with the experimental results of compound.

Response 3: The referenced statement refers to the docking scores. We have edited this statement to read “The docking scores of the O-methylated flavonoids (2-5) with MAO-A matched well with the trend in the experimental IC50, that the interactions are strong for all four of them but that the order of binding well is 1 better than 2 better than 3 better than 4.”

Thanks very much for your careful observation. In the referenced sentence, we made a mistake in the compound numbering for O-methylated flavonoids. The O-methylated compounds are 1 and 4-6. We fixed the numbering in this sentence in the abstract of the revised manuscript. The referenced statement refers to the binding free energy scores. We replaced the original “MAO-A formed strong and stable interactions with each of the O-methylated flavonoids (2-5).” with the following sentence: “The binding free energies of the O-methylated flavonoids (1 and 4-6) and chalcones (2-3) to MAO-A matched closely with the trend in the experimental IC50’s

Point 4: And, in my opinion, even the flavonoid scaffold was not list as a frequent hit or PAINS compound, they were reported existing in many kinds of plants extensively and to be active in many, many biological assays. They should be treated with special cautions, especially as lead compound for drug development.

Response 4: We added the cautionary statement about this in the revised manuscript. We added the following line in the conclusion section: “It is important to mention that the flavonoid scaffold possesses promiscuous biological activity that may be due to inherent structural features. For this reason, they should be treated with caution as lead compounds for drug development.”

Point 5: The manuscript should be edited carefully.

    In Figure 1, the illustration of ring A, B and C mentioned in the paper should be provided.

Response 5: The conventional flavonoid ring labels and numbering have been added to the image of compound 1 in in Figure 1 of the revised manuscript.

Point 6: Table 1 should be well prepared. For example, in the annotation of Table 1, it would be better if “b SI = [selectivity index]” was changed to “b SI means selectivity index, which was defined as MAO-A IC50/MAO-B IC50”.

Response 6: We have edited the table, including modifying the referenced footnote and moving “The results with significantly potent inhibition are presented in bold.” Into the figure caption. We also corrected the footnote letters used.

Point 7: The MAO-B IC50 of compound 5 does not need to be bold.

Response 7: We have made this edit.

Point 8: The symbol for Clorgyline and Deprenyl can be change to “c” since “b” was used for selectivity index.

Response 8: We have made this edit.

Point 9: The first line was not indented in part ‘4.2 MAO Inhibitory Assay’.

Response 9: We have made this edit.

Point 10: The first sentence in ‘4.7 Computational analysis’ lacked a closing bracket.

Response 10: We have made this edit.

Point 11: The word ‘manuscript’ in the last sentence of Author contributions didn’t need to be bold.

Response11: We have made this edit.

Point 12: The journal name of Reference 21 should be abbreviated.

Response 12: We have made this edit. Reference 21.

Reviewer 3 Report

In this work the authors isolated 6 flavonoid natural products and experimentally evaluated their inhibitory potency against both isoforms of the MAO enzymes. The binding of each ligand and their affinities were further rationalized through computational docking studies. The work is relevant for the pharmacology, since both MAOs are primarily clinical targets against various neurological diseases and the development and characterization of new and highly selective inhibitors is of prime interest. Although the obtained affinities are significantly lower than the reference compounds clorgyline and deprenyl, the obtained results demonstrate the potential of these organic skeletons as lead compounds for further development. As such, this work could be of a significant interest for a broad readership of Molecules, which makes the selection of this journal justified.

This multidisciplinary work is carefully planned and nicely executed. The choice of both computational and experimental approached is very appropriate for this kind of studies and the obtained results complement each other. The manuscript is nicely written and it reads well. In my opinion, this work is worth publishing in Molecules but only after it undergoes some revision and is reevaluated considering the following points:

- The authors demonstrated the potential of all six investigated compounds to act as reversible MAO inhibitors, yet their efficiency is gauged against reference drugs clorgyline and deprenyl, which are known irreversible inhibitors. This significant difference, together with the clinical relevance of highly selective reversible MAO inhibitors should be mentioned and discussed in the text, and supplemented with appropriate references.

- At several places the authors discuss the relevance of the O-methylation for modulating the binding affinities and demonstrate its beneficial effect for the inhibition potency (for example, 1 vs. 5, and 2 vs. 3). Literature already advises that the methylation is clearly beneficial for both the catalytic activity (Chemistry - A European Journal 2017, 23, 2915) and the irreversible inhibition of MAO enzymes (ACS Chemical Neuroscience 2019, 10, 3532) as such derivatives are obviously better positioned within the hydrophobic MAO active site, which affects their activity. These papers should be referenced and discussed in the text.

- The calculated Gibbs free binding energies are clearly overestimated, as these are reported between 28-76 kcal/mol in Table 3. Since the measured Ki values are at a micro-molar range, the binding affinities should be somewhere around 6-10 kcal/mol. This is a known limitation of the employed computations, yet this should be briefly mentioned in the text, especially for the benefit of casual readers. This is not a weak point of the current work, but the authors must emphasize this and underline the power of these computational techniques in relative terms among structurally similar ligands, which is the focus here.  

- The introductory discussion for the section 2.4. (pages 7-8) is largely technical in nature and should be placed at the end of the manuscript under the Materials and Methods section or Computational Details.

- The discussion in the manuscript would certainly be much easier to follow if the authors insert both the atom- and ring-labeling on Figure 1, which they lengthily use throughout the text.

- At several places in the text, the authors use a phrase "MAO receptor". I believe that a word "receptor" should be reserved for receptors, not the enzymes, while a more appropriate phrase would be "MAO binding site" or something similar.

- Page 10, line 257: I would certainly not term a 292-fold selectivity as "extrodinary". Even the reference drugs used here, deprenyl and clorgyline, show a much higher selectivity for the two MAO isoforms. Therefore, this should be rephrased.

- On a technical side, what were the protonation states of the employed ligands? Were these systems docked as neutral or charged compounds given their high acidity? Also, was MAO-B considered as a dimer or as a monomer just like MAO-A? These aspects should be mentioned in the section 4.7.

Author Response

Response to Reviewer 3 Comments

In this work the authors isolated 6 flavonoid natural products and experimentally evaluated their inhibitory potency against both isoforms of the MAO enzymes. The binding of each ligand and their affinities were further rationalized through computational docking studies. The work is relevant for the pharmacology, since both MAOs are primarily clinical targets against various neurological diseases and the development and characterization of new and highly selective inhibitors is of prime interest. Although the obtained affinities are significantly lower than the reference compounds clorgyline and deprenyl, the obtained results demonstrate the potential of these organic skeletons as lead compounds for further development. As such, this work could be of a significant interest for a broad readership of Molecules, which makes the selection of this journal justified.

This multidisciplinary work is carefully planned and nicely executed. The choice of both computational and experimental approached is very appropriate for this kind of studies and the obtained results complement each other. The manuscript is nicely written and it reads well. In my opinion, this work is worth publishing in Molecules but only after it undergoes some revision and is reevaluated considering the following points:

Point 1: The authors demonstrated the potential of all six investigated compounds to act as reversible MAO inhibitors, yet their efficiency is gauged against reference drugs clorgyline and deprenyl, which are known irreversible inhibitors. This significant difference, together with the clinical relevance of highly selective reversible MAO inhibitors should be mentioned and discussed in the text, and supplemented with appropriate references.

Response 1: We have added in discussion of these key points related to reversible vs. irreversible MAO inhibitors, and references about them.

Point 2: At several places the authors discuss the relevance of the O-methylation for modulating the binding affinities and demonstrate its beneficial effect for the inhibition potency (for example, 1 vs. 5, and 2 vs. 3). Literature already advises that the methylation is clearly beneficial for both the catalytic activity (Chemistry - A European Journal 2017, 23, 2915) and the irreversible inhibition of MAO enzymes (ACS Chemical Neuroscience 2019, 10, 3532) as such derivatives are obviously better positioned within the hydrophobic MAO active site, which affects their activity. These papers should be referenced and discussed in the text.

Response 2: We have added in reference to these papers and discussion of the important information and discussion they provide.

Point 3: The calculated Gibbs free binding energies are clearly overestimated, as these are reported between 28-76 kcal/mol in Table 3. Since the measured Ki values are at a micro-molar range, the binding affinities should be somewhere around 6-10 kcal/mol. This is a known limitation of the employed computations, yet this should be briefly mentioned in the text, especially for the benefit of casual readers. This is not a weak point of the current work, but the authors must emphasize this and underline the power of these computational techniques in relative terms among structurally similar ligands, which is the focus here. 

Response 3: We have added the requested suggestion in the revised manuscript, Section 2.4. The following sentence is included in the revised manuscript. “The calculated binding free energies vary between 28-76 kcal/mol against MAO-A and -B. Since some of the measured Ki values are in the micromolar range, the binding affinities should be somewhere around 6-10 kcal/mol. This is a known limitation of the employed computations, which are useful not on an absolute scale but in relative terms among structurally similar ligands, which is the focus here.”

Point 4: The introductory discussion for the section 2.4. (pages 7-8) is largely technical in nature and should be placed at the end of the manuscript under the Materials and Methods section or Computational Details.

Response 4:  We have moved this material to the section referenced.

Point 5: The discussion in the manuscript would certainly be much easier to follow if the authors insert both the atom- and ring-labeling on Figure 1, which they lengthily use throughout the text.

Response 5: > Figure 1 is now changed as suggested.

Point 6: At several places in the text, the authors use a phrase "MAO receptor". I believe that a word "receptor" should be reserved for receptors, not the enzymes, while a more appropriate phrase would be "MAO binding site" or something similar.

 Response 6:  We have edited this to remove the concept of “receptor” being attached to “MAO”.

Point 7: Page 10, line 257: I would certainly not term a 292-fold selectivity as "extrodinary". Even the reference drugs used here, deprenyl and clorgyline, show a much higher selectivity for the two MAO isoforms. Therefore, this should be rephrased.

Response 7: We have rewritten this to avoid exaggerating the significance of the determined selectivity.

Point 8: On a technical side, what were the protonation states of the employed ligands?

Response 8:  We have added the requested information in the revised manuscript. The predicted pKa (calculated by Epik module of the Schrödinger software) of the phenolic hydroxyl of studied flavonoids were found to be between 8 -9.5; therefore, at physiological pH 7.4, we used a neutral form of the molecules. We added in Section 4.7 “The compounds were docked as neutral molecules.”

Point 9: Were these systems docked as neutral or charged compounds given their high acidity?

Response 9: We have added the requested information in the revised manuscript. As mentioned above, we added in Section 4.7 “The compounds were docked as neutral molecules.”

Point 10: Also, was MAO-B considered as a dimer or as a monomer just like MAO-A? These aspects should be mentioned in the section 4.7.

Response 10: We have added into Section 4.7 that MAO-B was also considered as a monomer. The sentence reads “The protein structures of MAO-A and -B were each used as monomers in the docking study.”

Round 2

Reviewer 1 Report

From my point of view, the corrections were carried out successfully the manuscript can be accepted as it is

Reviewer 2 Report

Thanks for considering my suggestion.